# Finite–Discrete Element Method Simulation Study on Development of Water-Conducting Fractures in Fault-Bearing Roof under Repeated Mining of Extra-Thick Coal Seams

Longquan Mai and Hao Li *

Key Laboratory of In-Situ Property Improving Mining of Ministry of Education,
Taiyuan University of Technology, Taiyuan 030024, China; mlq925741532@163.com
* Correspondence: haolityut@163.com

**Abstract:** The formation of water-conducting fractures in overlying strata caused by underground coal mining not only leads to roof water inrush disasters, but also water-conducting fractures penetrate the aquifer, resulting in the occurrence of a mine-water-inrush disaster and the loss of water resources. It destroys the sustainability of surface water and underground aquifers. This phenomenon is particularly significant in extra-thick coal seams and fault-bearing areas. Numerical simulation is an effective method to predict the failure range of mining overburden rock with low cost and high efficiency. The key to its accuracy lies in a reasonable constitutive model and simulation program. In this study, considering that the three parts of penetrating cracks, non-penetrating cracks, and intact rock blocks are often formed after rock failure, the contact state criterion and shear friction relationship of discrete rock blocks and the mixed fracture displacement–damage–load relationship are established, respectively. Combined with the Mohr–Coulomb criterion, the constitutive model of mining rock mass deformation–discrete block motion and interaction is formed. On this basis, according to the engineering geological conditions of Yushupo Coal Mine, a numerical model for the development of water-conducting cracks in the roof with faults under repeated mining of extra-thick coal seams is established. The results show the following: The constitutive relation of the continuous deformation–discrete block interaction of overlying strata and the corresponding finite element–discrete element FDEM numerical program and VUSDFLD multi-coal seam continuous mining subroutine can numerically realize the formation process of faults and water flowing fractures in overlying strata under continuous mining of extra-thick multi-coal seams. The toughness of sand mudstone is low, and the fracture will be further developed under the repeated disturbance of multi-thick coal seam mining. Finally, it is stabilized at 216–226 m, and the ratio of fracture height to mining thickness is 14.1. When the working face advances to the fault, the stress concentration occurs in the fault and its overlying rock, which leads to the local fracture of the roof rock mass and the formation of cracks. The fault group makes this phenomenon more obvious. The results have been preliminarily applied and tested in Ningwu mining area, which provides theoretical support for further development of roof water disaster control under the condition of an extra-thick coal seam and avoids the loss of water resources in surface water and underground aquifers.

**Keywords:** extra thick; coal; faults; FDEM; fractures; VUSDFLD; aquifers; sustainability

## 1. Introduction

Coal is a major fuel used for generating electricity worldwide. It provides around 27% of the global primary energy need and generates about 36% of the world's electricity [1]. However, large-scale coal mining has led to a decline in groundwater levels [2,3] and increases in water pollution [4], vegetation death [5], land desertification [6], etc. Due to these concerns, investigating the development height of the water-conducting fractured zone (WCFZ), the primary channel through which phreatic water and underground water

enter the mining pit, is necessary for sustainable and coordinated development of mining areas and sustainable development of groundwater in mining areas [7,8].

The key to water-preserved mining is to maintain the normal water-resisting capacity of the aquiclude after mining [9], and the core lies in the development law of mining-induced fractures [10]. As a result of the in situ stress balance being disrupted by coal mining activities, the overlying rock layer undergoes deformation, resulting in the formation of water-conducting cracks [11]. Yang et al. [12] and Islam et al. [13] believe that the change in permeability of overlying strata caused by coal mining accelerates the loss of aquifer water and leads to the destruction of water resources. Based on experimental tests and a simulation analysis, Adhikary and Guo [14] and Wang et al. [15] proposed that the distribution and height of water-conducting fractures in a mining overburden are mainly affected by factors such as mining depth, coal seam thickness, mining speed, and key stratum position. Shabanimashcool and Li [16] and Suchowerska et al. [17] proposed a formula for estimating the height of water-conducting fractures. Vu, T. T [18], on the basis of an analysis and assessment of geological conditions, technological parameters, and support methods of a longwall, has pointed out the causes and rules of the phenomenon of face spall and roof falling in a fully mechanized longwall. Rak, Z et al. [19] present two typical courses of changes in the extractable resource mining rates. In order to protect aquifers in the upper part of coal seams, countries such as the United Kingdom, Japan, and Germany have formulated laws and regulations. However, the height of a water flowing fracture in overlying strata under the condition of an extra-thick coal seam, multi-coal seam, and fault is still unclear [10].

Numerical simulations are an effective and low-cost means to realize the distribution and evolution of water-conducting fractures in overlying strata under complex conditions with low cost and high efficiency [20]. The key to its accuracy lies in the establishment of a reasonable mechanical theory and the preparation of the corresponding numerical program [21]. At present, scholars mainly conduct numerical studies on water-conducting fracture height in mining overburden based on two basic assumptions. (1) Continuous medium assumption: Based on the Mohr–Coulomb yield criterion, Liu [22] and Wang et al. [23] used FLAC3D to establish an elastoplastic constitutive model, and combined it with in situ measurements to obtain the distribution shape and height of cracks in overlying rock strata. Thin et al. [24] used a double-yield constitutive model to simulate the pressure-bearing behavior of rock in the goaf, and obtained stress and crack distribution. Shabanimashcool et al. [16] used FLAC3D to simulate the progressive collapse and fracture process of the roof rock layer and the consolidation of the collapsed material.(2) Discrete medium assumption: Liu et al. [25] used UDEC to simulate the damage characteristics of overlying rock strata during fully mechanized caving mining; by combining borehole television and hydrological observation methods, they obtained the crack height and the ratio of crack height to mining thickness. With the 3DEC program, Cheng [26] studied the failure law of the overlying strata of the coal seam roof in the superposition mining of the working face at Wulunshan Coal Mine and obtained the height of the water-conducting fracture and the water inflow following mining. In addition, Azrag EA et al. [27] uses finite element simulation to study complex mine water problems. Hernández JH et al. [28] applied finite element thinking to flooding problems in an open-pit mine in the Balces River basin in La Coruña, Spain.

The above finite element method (FEM) and finite difference method (FDM) can only obtain the deformation and plastic zone of a mining overburden, but cannot obtain the results of a roof fracture and collapse. The discrete element method (DEM) is suitable for a roof with fully developed joints, and the joint or fractures are often arranged in the form of "block distribution", which easily leads to a large deviation between the numerical results and the measured results [29]. Under the fault condition of an extra-thick and multi-coal seam working face, the coal seam roof, especially the fault zone, is significantly transformed from a continuum into a discrete body with a random shape. More importantly, in this process, the quasi-brittle rock has a tensile-shear mixed ductile fracture, and the

structural plane has a separation/compression/shear friction coupling response. The above continuous–discrete transformation process is difficult to be solved only by FEM, FDM, or DEM [30–32]. In addition, in previous algorithms, most of the coal seams are regarded as discrete block groups. After reaching a certain time threshold, the discrete block groups are deleted instantaneously to achieve the purpose of working face advancement and coal seam mining. Meanwhile, coal seam mining is a continuous process of disturbance stress generation and release, and stress is the root cause of coal rock failure [33–35].

In order to solve the above problems, in this study, taking Yushupo Coal Mine as the engineering background, the mechanical constitutive relationship of the continuous deformation–fracture–discrete block interaction of a mining overburden was established, and the corresponding finite element–discrete element (FDEM) numerical program was formed. The VUSDFLD subroutine was also written to accurately determine the specific positional coordinates of the coal units and realize the continuous mining of coal over time. Combined with the in situ experiment of the flushing fluid consumption method, the development height of the water-conducting fractures of the mining overburden under the condition of the extra-thick multi-coal seams passing through faults was obtained. It lays a foundation for further research on water-preserved coal mining under extra-thick coal seams and complex geological conditions.

## 2. Constitutive Relation of Continuous Deformation—Discrete Motion of Rock

Under the mining influence, rock often transforms from a continuous medium to discrete medium. This mechanical process involves the elastic–plastic deformation and fracture of intact rock, as well as the separation, extrusion, and shear friction motion between broken blocks.

### 2.1. Elastic–Plastic Deformation of Intact Rock

In the numerical model, the solid element is used to represent the intact rock. Its mechanical properties include elastic and plastic deformation.

The elastic deformation is determined by the generalized Hooke's law [36], and the expression is

$$\sigma_{ij} = \lambda \varepsilon_{kk} \delta_{ij} + 2G\varepsilon_{ij} \tag{1}$$

where $\sigma_{ij}$ is the stress component, $\varepsilon_{ij}$ is the strain component ($i, j = 1, 2, 3$), $\lambda$ is the Lame constant, and $G$ is the shear modulus.

Once the stress reaches the Mohr–Coulomb criterion, the rock enters the plastic deformation stage, and the expression is

$$\tau = c + \sigma \tan \varphi \tag{2}$$

where $c$ is the cohesive force; $\varphi$ is the internal friction angle; and $\tau$ and $\sigma$ are the shear stress and the principal stress on the slip surface, respectively.

Plastic flow law:

$$d\varepsilon_{ij}^p = d\theta \frac{\partial \Phi}{\partial \sigma ij} \tag{3}$$

where $\varepsilon_{ij}^p$ is the plastic strain component, $\Phi$ is the plastic potential function, and $d\theta$ is the plastic factor. The expression of $\Phi$ is

$$\Phi = \left[ (\delta \sigma_t \tan \psi)^2 + (Bq)^2 \right]^{1/2} - p \tan \psi \tag{4}$$

where $q$ is the deviatoric stress; $p$ is the spherical stress; $\delta$ is the cusp curvature of the meridian of the plastic potential function in the $q$–$p$ plane in the tension section, and its

value is 0.1; $\psi$ is the dilation angle; and *B* controls the shape of the plastic potential function (*G*) on the $\pi$ plane and its expression is

$$B = \frac{4(1 - e^2)\cos^2\alpha + (2e - 1)^2 A(\pi/3)}{2(1 - e^2)\cos\alpha + (2e - 1)\cdot[4(1 - e^2)\cos^2\alpha + 5e^2 - 4e]^{1/2}} \tag{5}$$

where *e* is the eccentricity, with $e = (3 - \sin\alpha)/(3 + \sin\alpha)$; $\alpha$ is the polar angle.

### 2.2. Mixed Fracture of Non-Interpenetrated Cracks

In the numerical model, the cohesive elements were used to represent the potential cracks of rock mass. Under the complex stress state of tension and shear, the rock mass breaks from above elements. In the elastic deformation stage of a fracture, the relationship between traction force and separation displacement is

$$t = \left\{ \begin{array}{c} t_n \\ t_s \\ t_t \end{array} \right\} = \left[ \begin{array}{ccc} E_{nn} & 0 & 0 \\ 0 & E_{ss} & 0 \\ 0 & 0 & E_{tt} \end{array} \right] \left\{ \begin{array}{c} \delta_n \\ \delta_s \\ \delta_t \end{array} \right\} = \boldsymbol{E}\boldsymbol{\delta} \tag{6}$$

where $t_n$, $t_s$, and $t_t$ are the loads in the normal direction and the two tangential directions of the cohesive element, respectively; $E_{nn}$, $E_{ss}$, and $E_{tt}$ are the stiffnesses in the normal direction and the two tangential directions, respectively; and $\delta_n, \delta_s, \delta_t$ are the displacements in the normal direction and the two tangential directions, respectively.

The cohesive element begins to fracture when the stress meets the following conditions:

$$\max\left\{ \frac{\langle t_n \rangle}{t_n{}^0}, \quad \frac{t_s}{t_s{}^0}, \quad \frac{t_t}{t_t{}^0} \right\} = 1 \tag{7}$$

where $t_n{}^0$, $t_s{}^0$, and $t_t{}^0$ are the peak loads in the normal direction and the two tangential directions of the element, respectively; $\langle t_n \rangle$ indicates that only the tensile load is considered in the normal direction of the element, and when $t_n < 0$, it is a compressive load, while $\langle t_n \rangle = 0$ indicates that the compressive stress has no effect on the fracture.

As the separation displacement continues to increase, the area of the contact surface between the rock blocks on both sides of the crack tip gradually decreases, resulting in a decrease in tensile and shear strength. For this reason, damage is introduced, and the correlation between mixed fracture displacement and damage variables is established.

$$D = \frac{\delta_m^f(\delta_m^{\max} - \delta_m^0)}{\delta_m^{\max}(\delta_m^f - \delta_m^0)} \tag{8}$$

where *D* is the damage variable; $\delta_m^{\max}$ is the mixed total displacement of the crack tip, and its value is the vector sum of the pure tensile and pure shear displacements; and $\delta_m^0$ is the total displacement when the cohesive element initially fractures.

The tensile and shear strengths of cohesive elements after damage are

$$\begin{array}{l} tn = \left\{ \begin{array}{l} (1 - D)\bar{t}n, \bar{t}n \geq 0 \text{ Tensile stress} \\ \bar{t}n \text{ Compressive stress does not cause damage} \end{array} \right. \\ t_s = (1 - D)\bar{t}_s, t_t = (1 - D)\bar{t}_t \end{array} \tag{9}$$

Whether cohesive elements are completely fractured can be ascertained according to the fracture critical displacement under tensile and shear loads.

$$\delta_F = \sqrt{\langle \delta_n \rangle^2 + \delta_s{}^2 + \delta_t{}^2} \tag{10}$$

### 2.3. Mechanical Response of Penetrating Cracks

Interactions among discrete rock blocks can be calculated by the discrete element method (DEM). In the numerical calculation, according to the block contact and relative

motion trends, the relationship between two blocks falls into one of three categories, i.e., separation, compression, and shear friction, respectively, detailed as follows:

(1)  Separation occurs when any node spacing (*l*) between adjacent solid elements (simulating intact-rock blocks) is more than 0. At this time, Newton's second law is used to calculate the rock movement under the influence of gravity [37].

(2)  If *l* = 0 and there is compressive stress between adjacent rock blocks, penetrating cracks undergo a compression extrusion process.

(3)  If *l* = 0 and there is shear stress along the penetrating crack, shear friction occurs.

When the adjacent blocks squeeze each other, the constitutive relation is

$$\sigma_n = \frac{D_n N_{\max} n}{N_{\max} - n} \tag{11}$$

where $\sigma_n$ is the compressive stress; $N_{\max}$ is the maximum closure of the structural surface, a constant parameter determined with three-dimensional shape scanning experiments; $n$ is the closure of the structural surface; and $D_n$ is the normal modulus of the structural surface. In this study, the values are the parameters of the corresponding adjacent rock blocks.

When shear friction occurs, based on the experimental data and research results [38], the constitutive relation is

$$\sigma_s = \frac{D_s S + \sigma_{s,p}(p-1)(S/s_p)^2}{1 + (D_s \cdot s_p/\sigma_{s,p} - 2)(S/s_p) + D_s(S/s_p)^2} \tag{12}$$

where $\sigma_s(\sigma_{s,p})$ is the (peak) shear stress, $S(s_p)$ is the (peak) shear displacement, and $D_s$ is the shear stiffness.

## 3. Material Parameters and Verification of Constitutive Equations

The material parameters of the above constitutive equations were obtained by fracture mechanics experiments and direct shear experiments. Constitutive equations were verified by comparing the results of a numerical simulation and laboratory experiment.

### 3.1. Material Parameters

The mechanical parameters of the rock matrix were obtained according to the literature [37]. The parameters of a tension and shear mode fracture were obtained based on experiments of three-point bending and punch-through shear under loading and unloading conditions (Tables 1 and 2). Shear friction parameters of the structural plane were obtained with the direct shear test. According to the mining depth of the Yushupo Coal Mine, the normal stress ($\sigma_n$) values of 1, 6, and 7 MPa were determined, and the shear friction parameters were obtained for different $\sigma_n$.

**Table 1.** Mechanical parameters.

| | $\rho/(\text{kg}\cdot\text{m}^{-3})$ | $t_s^0, t_t^0/\text{MPa}$ | $t_n^0/\text{MPa}$ | $E_{nn}/\text{GPa}$ | $E_{ss}, E_{tt}/\text{GPa}$ | $\delta_{max}^m/m$ | $E/\text{GPa}$ | $\mu$ |
|---|---|---|---|---|---|---|---|---|
| Fault gouge | 1500 | 10 | 1.5 | 0.2 | 2.53 | 0.17 | 6.52 | 0.35 |
| Siltstone | 2500 | 7.2 | 3.6 | 4 | 16.23 | 0.06 | 17.23 | 0.22 |
| Limestone | 2500 | 20 | 5.7 | 6.2 | 37.28 | 0.04 | 41.56 | 0.25 |
| Coal | 1500 | 10 | 3.5 | 3 | 3.98 | 0.06 | 10.21 | 0.25 |
| Mudstone | 1500 | 3.5 | 2 | 3 | 3.56 | 0.12 | 11.85 | 0.25 |
| Sandy mudstone | 2500 | 10 | 3.5 | 15 | 6.71 | 0.07 | 15.33 | 0.25 |

**Table 1.** *Cont.*

| | $\rho/(\text{kg} \cdot \text{m}^{-3})$ | $t_s{}^0, t_t{}^0/\text{MPa}$ | $t_n{}^0/\text{MPa}$ | $E_{nn}/\text{GPa}$ | $E_{ss}, E_{tt}/\text{GPa}$ | $\delta_{max}^m/m$ | $E/\text{GPa}$ | $\mu$ |
|---|---|---|---|---|---|---|---|---|
| Bauxite mudstone | 2500 | 10 | 3.5 | 15 | 4.53 | 0.09 | 13.86 | 0.25 |
| | $D_s/\text{GPa}$ | $\sigma_s/\text{MPa}$ | $\sigma_r/\text{MPa}$ | $S_r/\text{mm}$ | $S_p/\text{mm}$ | $\varphi/(°)$ | $c/\text{MPa}$ | $\psi/(°)$ |
| Fault gouge | 0.26 (1) 0.96 (6) 1.03 (7) | 0.69 (1) 2.25 (6) 2.72 (7) | 0.38 (1) 1.44 (6) 1.91 (7) | 7.64 (1) 9.11 (6) 9.03 (7) | 1.89 (1) 2.54 (6) 2.66 (7) | 27 | 3.73 | 22 |
| Siltstone | 2.04 (1) 10.40 (6) 11.06 (7) | 1.47 (1) 8.40 (6) 9.49 (7) | 0.82 (1) 4.55 (6) 5.13 (7) | 2.63 (1) 6.18 (6) 6.30 (7) | 0.72 (1) 0.79 (6) 0.84 (7) | 40 | 12.35 | 34 |
| Limestone | 1.96 (1) 9.21 (6) 10.11 (7) | 1.21 (1) 7.26 (6) 8.21 (7) | 0.62 (1) 4.12 (6) 4.67 (7) | 1.94 (1) 5.81 (6) 5.93 (7) | 0.63 (1) 0.65 (6) 0.69 (7) | 45 | 22.70 | 34 |
| Coal | 0.68 (1) 3.67 (6) 4.27 (7) | 0.72 (1) 4.11 (6) 4.79 (7) | 0.34 (1) 2.22 (6) 2.59 (7) | 5.61 (1) 8.54 (6) 9.13 (7) | 1.06 (1) 1.14 (6) 1.16 (7) | 30 | 4.77 | 25 |
| Mudstone | 0.37 (1) 1.03 (6) 1.12 (7) | 0.89 (1) 3.08 (6) 3.44 (7) | 0.47 (1) 1.86 (6) 2.09 (7) | 8.50 (1) 9.58 (6) 9.54 (7) | 2.43 (1) 2.96 (6) 3.05 (7) | 32 | 7.36 | 26 |
| Sandy mudstone | 0.57 (1) 1.45 (6) 1.56 (7) | 1.35 (1) 3.66 (6) 4.13 (7) | 0.86 (1) 2.36 (6) 2.68 (7) | 9.88 (1) 10.55 (6) 10.92 (7) | 3.23 (1) 4.12 (6) 4.23 (7) | 38 | 9.88 | 32 |
| Bauxite mudstone | 0.45 (1) 1.33 (6) 1.42 (7) | 1.11 (1) 3.52 (6) 3.92 (7) | 0.65 (1) 2.11 (6) 2.44 (7) | 9.12 (1) 10.22 (6) 10.45 (7) | 2.88 (1) 3.16 (6) 3.23 (7) | 38 | 9.13 | 30 |

**Table 2.** Statistical crack width-related data.

| Mining Distance (m) | Cracks Opening at Measuring Line 1 (m) | Cracks Opening at Measuring Line 2 (m) | Cracks Opening at Measuring Line 3 (m) | Cracks Opening at Measuring Line 4 (m) | Cracks Opening at Measuring Line 5 (m) | Cracks Opening at Measuring Line 6 (m) |
|---|---|---|---|---|---|---|
| 0 | 14.61 | 1.6 | 1.21 | 1.6 | 4.52 | 1.43 |
| 53.7 | 15.2 | 1.94 | 1.23 | 1.6 | 4.67 | 1.42 |
| 73.45 | 21.24 | 1.8 | 1.56 | 1.71 | 5.13 | 1.36 |
| 211.9 | 24.14 | 2.5 | 3.4 | 5.7 | 4.98 | 1.62 |
| 282.5 | 21.62 | 2.53 | 2.43 | 4.97 | 7.64 | 1.7 |
| 353.1 | 22.39 | 5.56 | 3.74 | 5 | 5.63 | 2.2 |
| 423.75 | 24.81 | 9.26 | 4.56 | 4.92 | 6.6 | 2.5 |
| 494.4 | 23.94 | 9.72 | 5.25 | 5.87 | 8.58 | 2.6 |
| 558.4 | 26.77 | 11.55 | 7.85 | 7.55 | 8.45 | 3.67 |

### 3.2. Constitutive Equation Verification

The validity and accuracy of the constitutive equations for different materials were verified with three-point bending, punch-through shear, and rough surface shear tests and simulations of offset prefabricated cracks.

The mixed fracture constitutive equations can be verified by comparing the experimental and simulation results. According to the size and boundary conditions of the specimen in the experiments (Figure 1a,b), the numerical model was established (Figure 2a). The displacement loading rate is 0.02 mm/min. Cohesive elements were arranged at the prefabricated crack, and solid elements were arranged at the remaining positions. The mechanical properties of elements are shown in Equations (1)–(10), and the mechanical parameters are shown in Table 1. In the table, ρ is the density of the coal matrix, and

parameters such as DS, Sp, σr, and Sr are all functions of the normal stress σn and the structural surface roughness coefficient JRC [38], and their values are obtained based on the experiment shown in Figure 1 The remaining parameters are explained in Section 2.

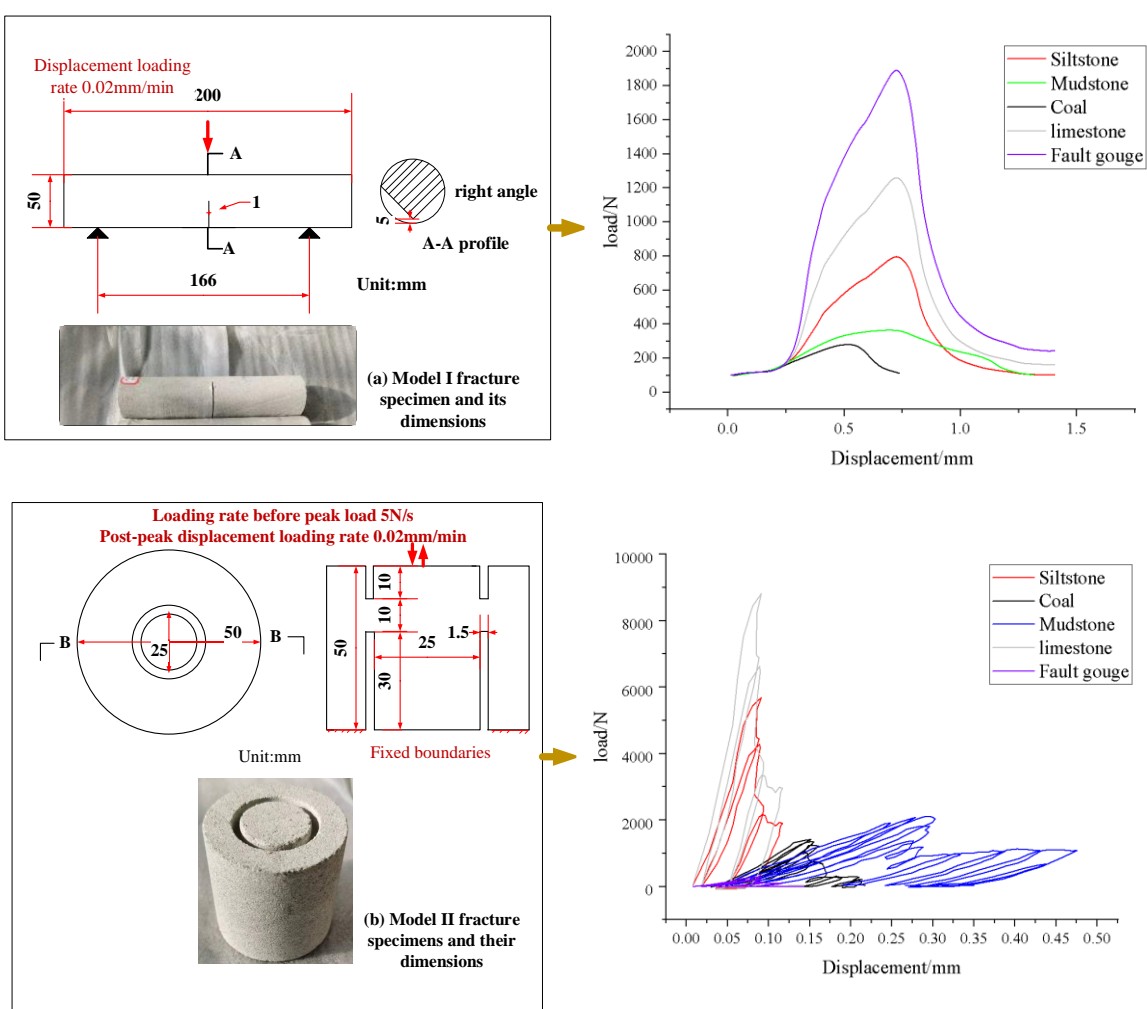

**Figure 1.** Experiments and load–displacement curves of mode I and II fracture mechanics.

For the direct shear test, the specimens with rough structural planes were prepared using the Brazilian splitting method. Considering that the mechanical parameters in Table 1 were obtained according to the normal stress $\sigma_n$ =1, 6, and 7 MPa, in order to verify the rationality of the constitutive model, $\sigma_n$ = 3 and 8 MPa was set, the lower plate was fixed, and the shear loading rate of the upper plate was 0.02 mm/min. When the data tended to be stable, the loading was stopped. The corresponding numerical calculation model was established, and the normal and shear friction contact properties were given at the Thanks for your check, I have modified the picturestructural plane (Equations (11) and (12)). The solid elements were arranged at other positions, and the mechanical parameters were determined using Table 1 and Figure 1 Through the experiment and FDEM calculation program (Figure 3), the experimental and simulation results are shown in Figure 2.

The following can be seen from Figure 2:

(1)  The load–displacement curves of tension and shear obtained from the numerical simulations were highly consistent with the experimental results. Taking siltstone as an example, the peak loads obtained from experiments (simulations) were 785 (770 N) and 5720 (5688 N), with simulation errors of 3% and 0.5%. The peak displacements were 0.74 (0.77) mm and 0.09 (0.10) mm, respectively, with simulation errors of 4% and

11%. According to the above comparison, the applicability of the fracture constitutive equations was verified.

(2) The mechanical parameters obtained from the experiments with $\sigma_n$ =1, 6, and 7 MPa were substituted into the constitutive equations of penetrating cracks; the shear stress–displacement curves under $\sigma_n$ = 3 and 8 MPa conditions were obtained. When $\sigma_n$ was increased from 3 MPa to 8 MPa, the peak shear stresses of the structural plane obtained from the experiment (simulation) were 4.82 (4.77) MPa and 10.68 (10.94) MPa, and the peak displacements were 0.89 (0.93) mm and 1.01 (1.10) mm, respectively. The above data show that the simulation results are close to the experimental results, indicating the rationality of the constitutive model.

(3) It should be noted that there is a nonlinear segment at the beginning of the curve obtained from the numerical simulation shown in Figure 2b. This is due to the fact that a 5 min linear loading phase was set up in the pre-displacement loading stage of the numerical model, and the loading rate was gradually increased from 0 to 0.002 mm/min to avoid the formation of shock loads.

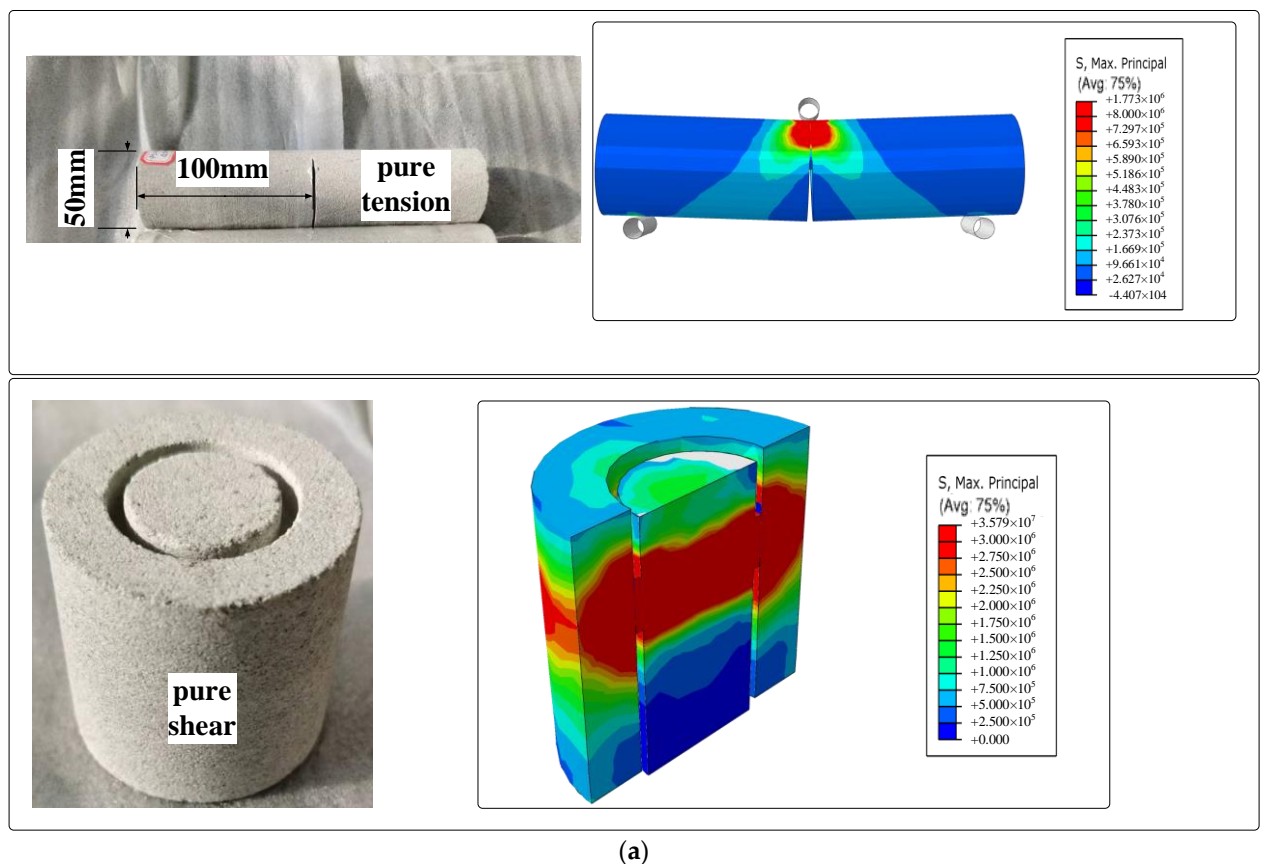

(**a**)

**Figure 2.** *Cont.*

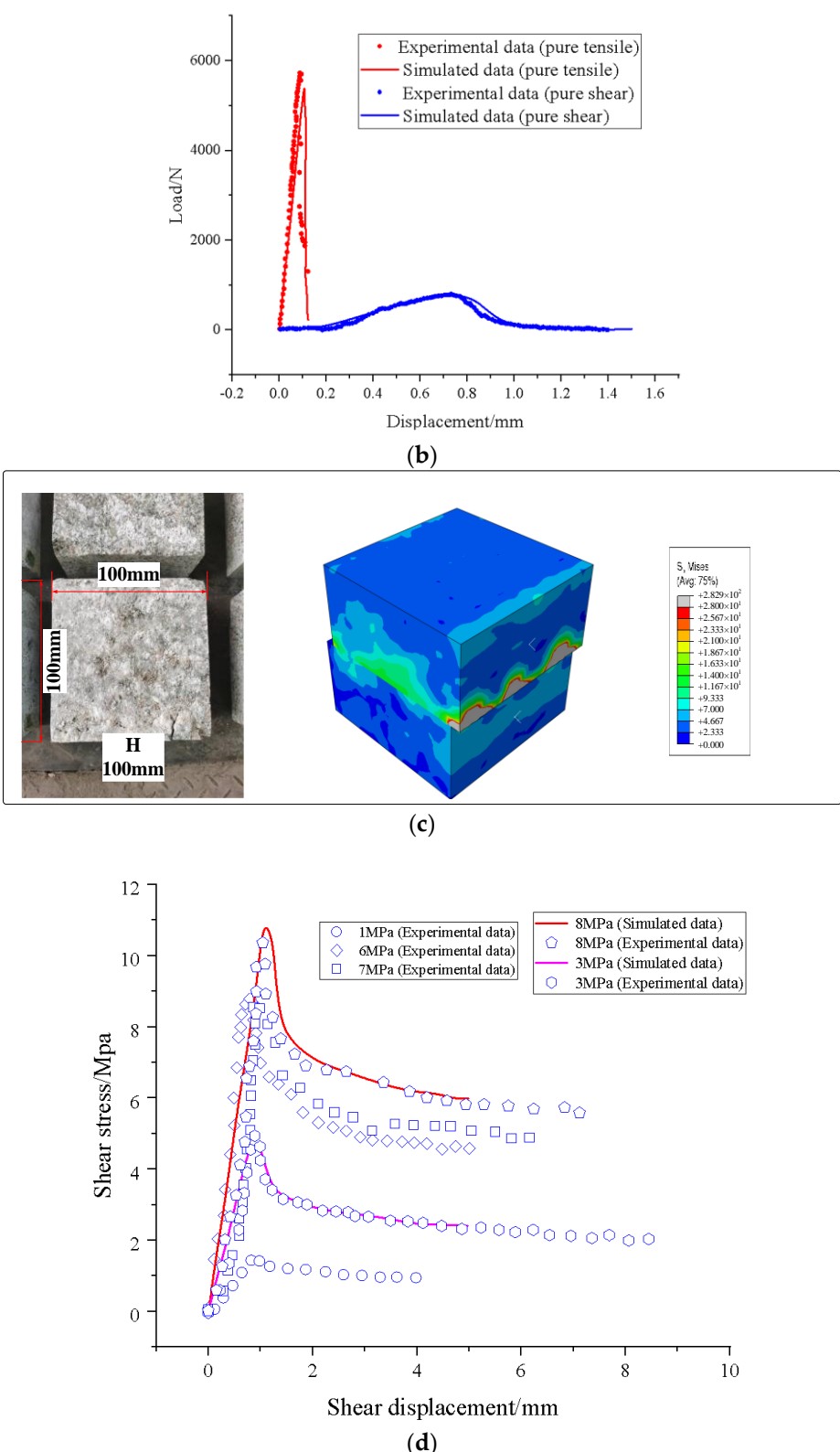

**Figure 2.** Rock fracture and shear friction experiments and numerical simulation results. (**a**) Experimental specimens and numerical models of mode I and II fractures. (**b**) Load–displacement curves. (**c**) Experimental specimen and numerical model of shear friction. (**d**) Results from shear friction experiments and numerical calculations.

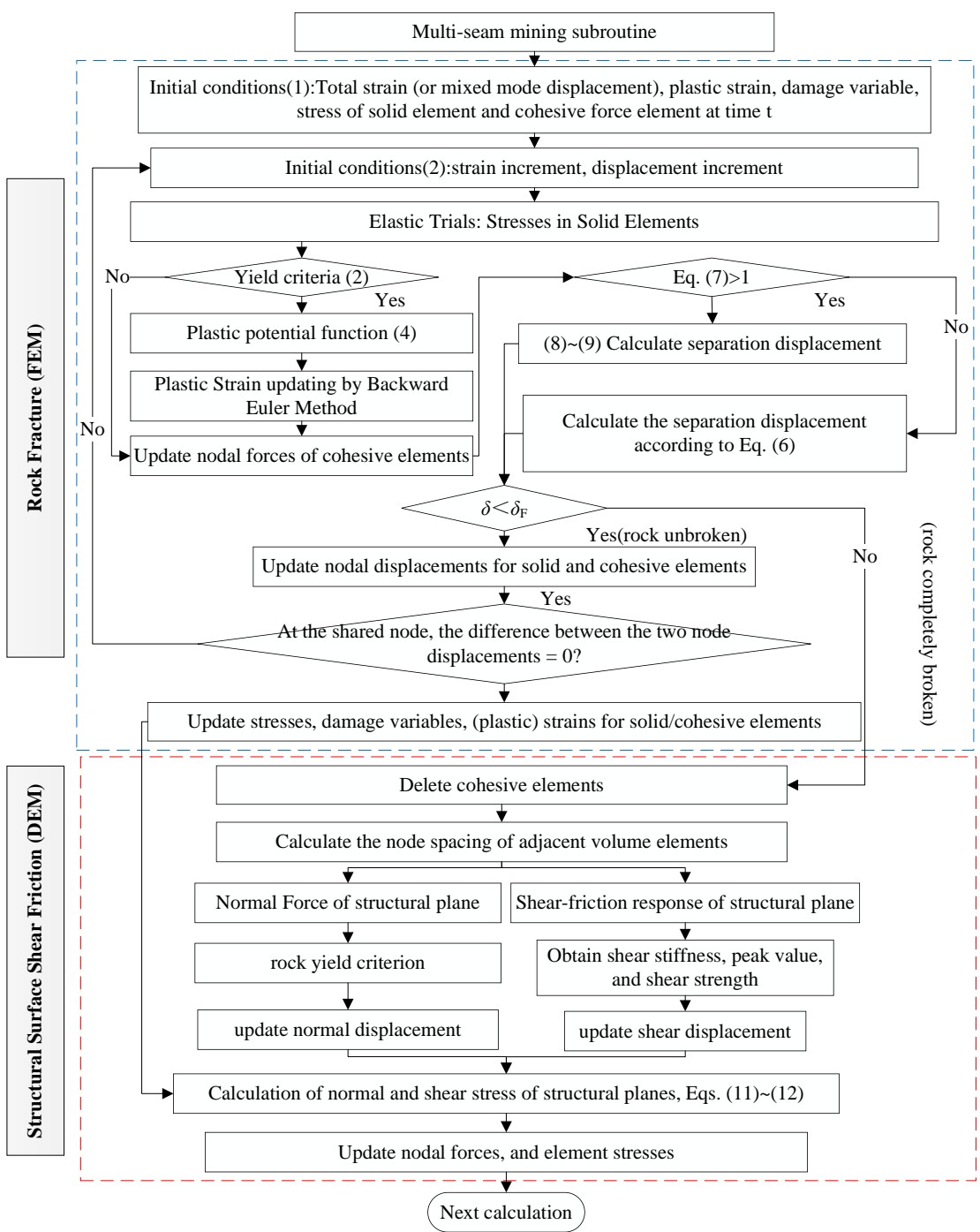

**Figure 3.** FDEM numerical calculation process.

## 4. Numerical Simulation Methods

### 4.1. FDEM Numerical Process

Based on the above-mentioned constitutive relations between the continuous deformation and discrete motion of rock mass, the corresponding finite–discrete element method (FDEM) numerical program was compiled to model the transformation process of a mining overburden from a continuous medium to discrete medium. The FDEM numerical calculation process is shown in Figure 3.

The fundamental concept of the FDEM numerical program is as follows: Initially, coal seams are mined through the VUSDFLD subroutine. During the mining process, the total strain (or mixed-mode displacement), plastic strain, damage variables, and stress of the

solid elements and cohesive elements at time t are obtained, thereby yielding the strain increment and displacement increment at this time. Subsequently, an elastic trial stress calculation for the solid elements is conducted. According to Equation (2), it is determined whether the stress satisfies the Mohr–Coulomb criterion. If it does, the plastic potential function is calculated according to Equation (4), and the plastic strain is updated using the backward Euler method, along with the nodal forces of the cohesive elements. If it does not satisfy the criterion, the nodal forces of the cohesive elements are updated directly. Equation (7) is then used to judge whether the cohesive elements begin to fracture. If they do, the separation displacement is calculated using Equations (8) and (9); otherwise, it is calculated using Equation (6). Subsequently, it is determined whether the separation displacement reaches the critical fracture displacement ($\sigma^F$) under tensile and shear loads. If $\sigma^F$ is not reached, the rock has not fractured at this time, and the nodal displacements of the solid and cohesive elements are updated. It is then checked whether the difference in nodal displacements at the shared nodes is zero. If it is, the stress, damage variables, and (plastic) strain of the solid or cohesive elements are updated; otherwise, the computation returns to the starting point. If the separation displacement reaches $\sigma^F$, the computation transitions to the discrete element method (DEM) module. Through the command status = 0, the cohesive elements are deleted, and the nodal distances of the adjacent solid elements are calculated, with normal and shear displacements updated accordingly. The normal compression and shear friction of the discrete rock structural surfaces are then calculated using Equations (11) and (12), nodal forces are updated, the stress of the elements is calculated using the finite element method, and finally, the next step of computation is performed.

### 4.2. Numerical Subroutine for Continuous Coal Mining

The above-mentioned FDEM program cannot automatically model the process of "coal seam mining, entity, and cohesive element deletion". To achieve the above purpose, the VUSDFLD subroutine was compiled based on FORTRAN to model the continuous mining of multiple coal seams. The execution flow chart of the algorithm is shown Figure 4 below.

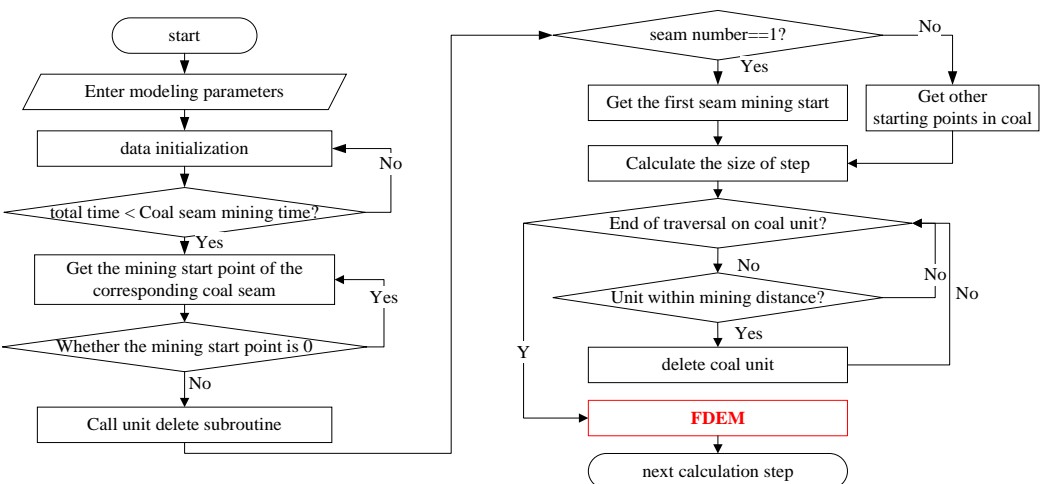

**Figure 4.** Continuous multi-seam mining procedure.

The basic approach of the VUSDFLD program is as follows:

(a) The establishment of numerical calculation models according to engineering geological conditions.

(b) The selection of the VUSDFLD subroutine as the interface. The unit data of the coal seam to be mined are extracted. If the advancing speed of the working face is speed m, according to the return parameters and the speed of the subroutine in the advancing

distance formula, the advancing distance can be obtained, and the time–displacement relationship is

$$\begin{cases} fstep = startPoint + totalTime \times speed \\ fstep = startPoint + (totalTime \\ \qquad -partCoalTime(step-1)) \times speed \end{cases} \tag{13}$$

where $fstep$ is the advancing distance, $startPoint$ is the starting point of mining, $totalTime$ is the total time of the return parameters, $speed$ is the speed, $step$ is the number of the coal seam, and $partCoalTime$ is the mining time corresponding to each coal seam.

The advancing distance is compared with the current integration point $X$ coordinates, and the characteristic length of the unit is added as a correction parameter.

$$coordMp(k,1) + charLength(k) <= fstep \tag{14}$$

where $k$ is the integration number; $coordMp(k,1)$ is the abscissa of the integration point; and $charLength$ is the characteristic length of the unit, which is used as a correction parameter.

When the above conditions are met, the corresponding unit state parameter is set to 0, and the unit is deleted. The above-mentioned process can be repeated for each following coal seam, thus allowing for the modeling of the continuous mining of multiple coal seams.

## 5. Simulation of Development of Overlying Rock Rupture Zone across Double Fault

### 5.1. Study Area

Yushupo Coal Mine is located in South Pingshuo of National Planning Mining Area in Ningwu Coalfield. The main coal-bearing strata are Carboniferous Taiyuan Formation and Permian Shanxi Formation. The roof aquifer in the minefield mainly includes the Quaternary pore aquifer and the bedrock weathering fissure aquifer, followed by the Permian sandstone fissure aquifer and the Upper Carboniferous Taiyuan Formation sandstone fissure aquifer (Figure 5). The average thickness of coal seam 2# is 6 m, the main roof is sandstone, the immediate roof is sandy mudstone, and the floor is mudstone. The 44 m below 2 # coal is a 5 # coal seam, with an average thickness of 16 m. The immediate roof is mudstone, and the main roof is siltstone. The mining depth of the 5 # coal seam is about 305 m. Due to the stress induced by the repeated and continuous coal mining, especially at faults, the mining fissure formed in the overlying rock strata may communicate with the pore aquifer in the Quaternary loose layer and the weathered-bedrock layer, representing a significant safety concern for the mining of an extra-thick coal seam.

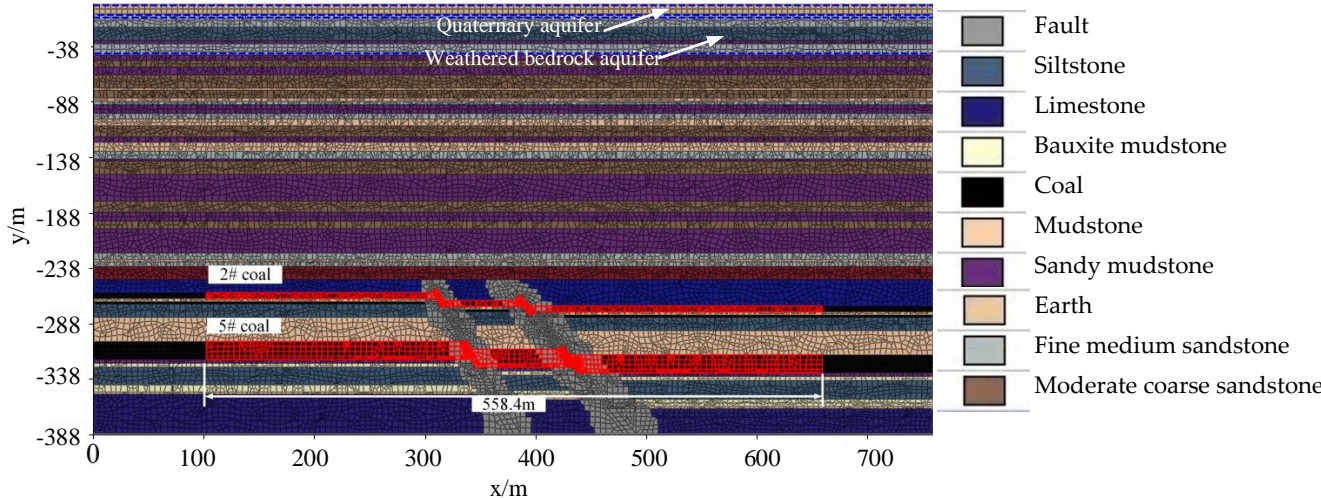

**Figure 5.** Numerical calculation model.

*5.2. Numerical Model*

In order to study the evolution law of water-conducting fractures in overburden strata in conditions of extra-thick multi-coal seams and faults, a numerical calculation model as shown in Figure 5 was established according to the engineering geological conditions of Yushupo Coal Mine. Considering the high nonlinearity of the constitutive equations, the size of the model was controlled to be x = 760 m (length) and y = 388 m (height). Normal constraints were applied at the left and right boundaries, bottom boundaries, and front and rear boundaries of the model, and self-weight loads were set in the y-axis direction. Intact coal rock blocks were represented by hexahedral solid elements, corresponding to the elastoplastic constitutive model. The potential fracture surfaces of the rock mass were represented by cohesive elements, corresponding to the fracture constitutive model. Once rock mass was completely fractured, the structural plane formed a contact pair, where the given interaction was constitutive of discrete rock masses. The rock mechanical parameters are shown in Table 1. The two red areas in the model are working faces of 2# and 5# coal seams, with a mining length of 558.4 m. Starting 100 m from the left boundary of the model, Coal Seam No. 2 is mined from left to right at a rate of 5 m per day, continuing until mining ends 100 m from the right boundary of the model. After completing the mining of Coal Seam No. 2, Coal Seam No. 5 is mined in the same manner.

The distribution of natural structural planes in the surrounding rock is in the form of Voronoi polyhedra [39], and the spacing is determined as follows: Considering the periodic collapse step distance of the sandy mudstone roof in the No. 5 coal working face and the tensile strength ratios of various rocks to sandy mudstone, the average spacing of natural structural planes in different rock layers along the working face advancement direction is determined to be 25.7 m for limestone, 15.2 m for sandy mudstone, 17.8 m for sandstone, 11.2 m for mudstone, and 1.5 m for fault zones. Solid elements are divided between natural structural planes, with the boundaries of these elements serving as potential fracture surfaces. The spacing of these potential fracture surfaces is determined based on the average core length ratios of various rocks, taking into account computational costs. The determined spacings are 6.1 m for limestone, 4.8 m for sandy mudstone, 5.5 m for sandstone, 4 m for mudstone, and 0.6 m for fault zones.

*5.3. Simulation Results*

Through the above numerical calculation model, the results of an overburden fracture after 2 # coal seam mining are obtained (Figure 6). With the advance of the working face, the suspended area of the lower part of the roof increases, the roof breaks and collapses, and the overlying rock fissures are formed. At this time, floor fractures of the 2 # coal seam have reached the 5 # coal seam (white line in Figure 6).

To study the propagation of water-conducting fissures in the overlying rock layer during the mining of extra-thick coal seam 5#, six survey lines (white dotted lines in Figure 6) were arranged with the vertical distances of 56 m, 96 m, 136 m, 176 m, 216 m, and 256 m from coal seam 5#. The water-conducting fractures with an opening of more than 1 mm were counted [39]. The cracks opening with the advancing distance at each measuring line are shown in Table 2.

According to the water-conducting fracture opening data of survey line 1 (Table 2), survey line 1 is located at the floor of 2 # coal sandstone and mudstone. Due to the mining influence of 2 # coal, a certain width of the water-conducting fracture had been developed in this position. Under the repeated disturbance of 5 # coal, the fracture opening further increased, resulting in a greater change in the results of line 1. For the convenience of observation, the total opening data from line 1 to line 6 were drawn into Figure 7, and the total opening data from line 2 to line 6 were drawn into Figure 8.

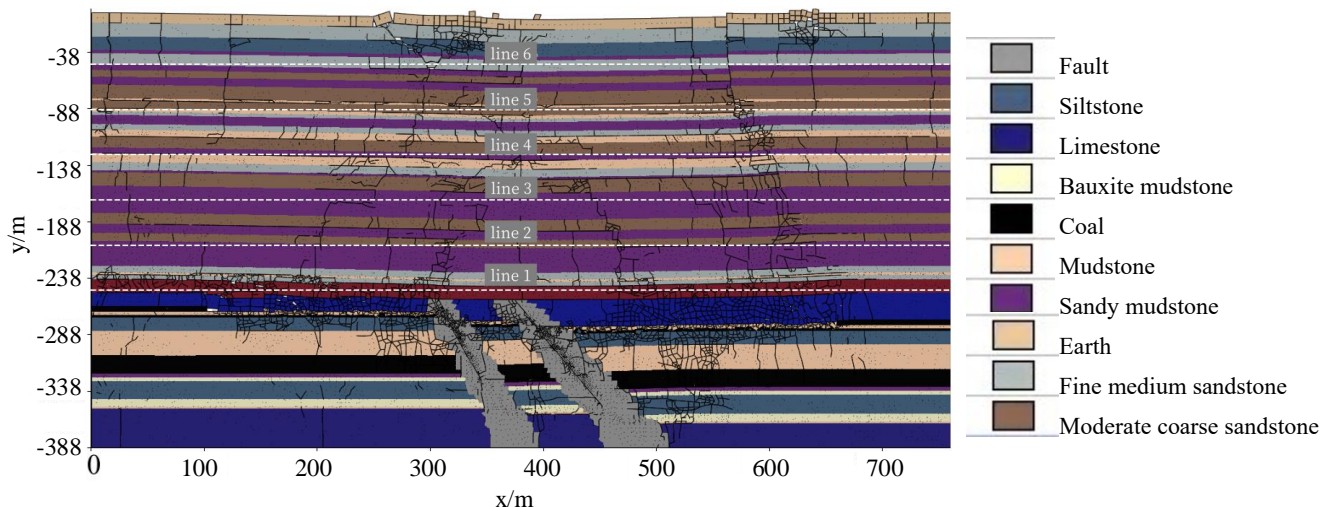

**Figure 6.** Survey-line position and fracture propagation after mining of coal seam 2 #.

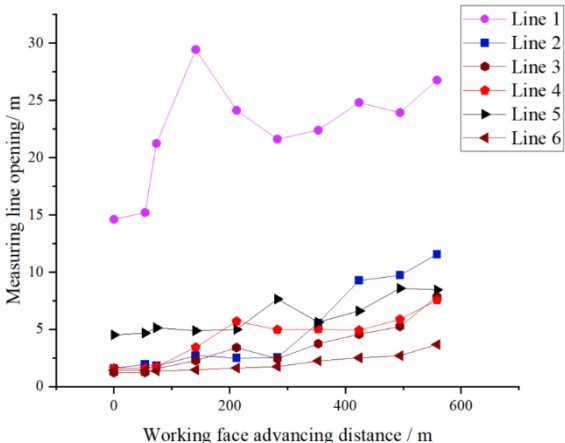

**Figure 7.** Crack opening at measuring lines 1~6.

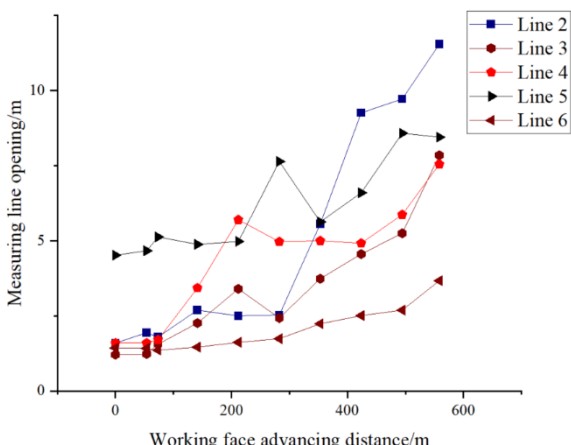

**Figure 8.** Crack opening at measuring lines 2~6.

It can be seen from Figures 7 and 8 that the opening of the water-conducting fractures at measuring lines 1 to 5 was characterized by a fluctuating trend with the working face advanced, showing an initial increase, followed by a decline, and finally another increase. Specifically, the opening in survey line 1 increased from 14.61 m (1.6 m from survey line 2) to 29.45 m (2.69 m), then decreased to 22.39 m (2.53 m), and finally increased to 26.77 m

(11.55 m). As shown in Figure 9, when the working face advanced to 53.7 m, the roof fractured for the first time and began to collapse. At this time, the rock formation was subjected to low stress and showed a slight growth trend. When the working face advanced to 73.45 m, the roof fractured completely and collapsed, and the goaf was filled with discrete and fractured rock blocks, causing periodic pressure. The width of the water-conducting fissure changed greatly and rapidly increased to 21.24 m (survey line 1). At the same time, the working face advanced to 141.25 m. After survey line 1 reached its peak at 29.45 m, the mining stress gradually stabilized as the working face advanced to 282.5 m, and the rock strata showed a general downward trend. At this time, the width of the fissure dropped to 21.62 m. When the working face advanced to 558.4 m, due to the large mining span, the roof collapsed and subsided severely, and the cracks increased in width.

According to Figure 6, lines 1, 2, 3, and 4 were located in the sandy mudstone layer; line 6 was located in the medium–coarse sandstone layer; and line 5 was located in the mudstone layer. From Figure 6 to Figure 8, it can be seen that at the survey lines in the sandy mudstone layer, the width and density of fissures were relatively greater than those in other formations, making the change in fissure width more evident. This is because the rock mass is not only subjected to high mining stress, but also has a lower fracture energy of sandy mudstone, which is prone to fragmentation, collapse, and the formation of cracks. After the mining of the 2# coal seam, there was a certain impact on the whole rock layer, and fracturing occurred in some places. Even though the medium–coarse sandstone layer and mudstone are characterized by high fracture energy, the movement and subsidence of the overburden rock during the advancement of the working face led to a further increase in fissure width in these rock layers.

Data of line 6 showed a slightly increasing overall trend in terms of crack width due to the overall movement of the overburden rock mass and the impact of subsidence. However, given that the overall change was not significant, it was inferred that the developed fissures did not reach measurement line 6. Therefore, the survey line was encrypted between survey line 5 and 6, and the distance between lines *a* and *b* and 5 # coal is 226 m and 236 m, respectively. The results are shown in Figure 10. It can be seen that the height of the final hydraulic fracture was in the interval of 216 m~226 m.

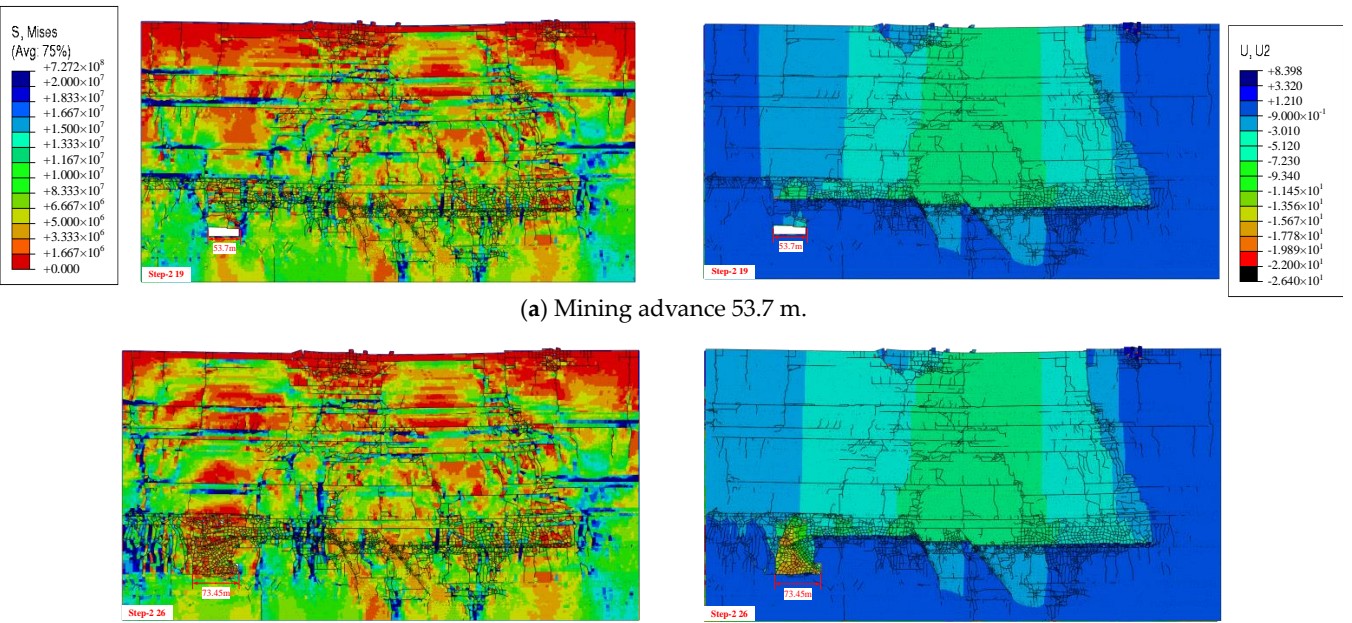

(**a**) Mining advance 53.7 m.

(**b**) Mining advance 73.45 m.

**Figure 9.** *Cont*.

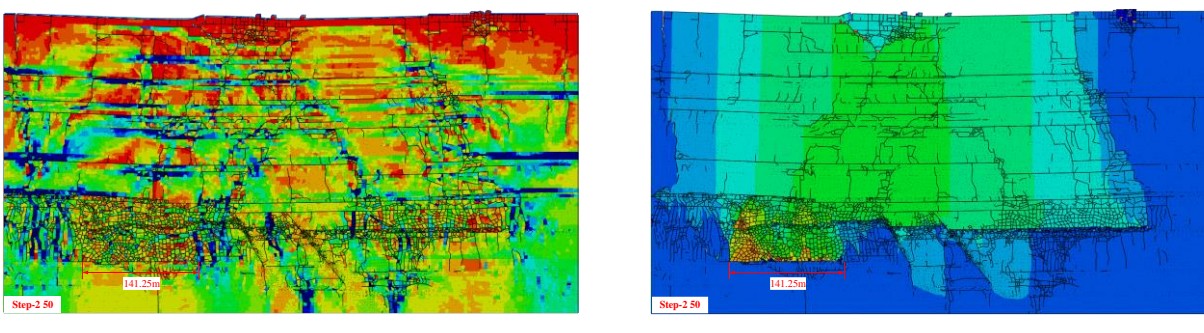

(**c**) Mining advance 141.25 m.

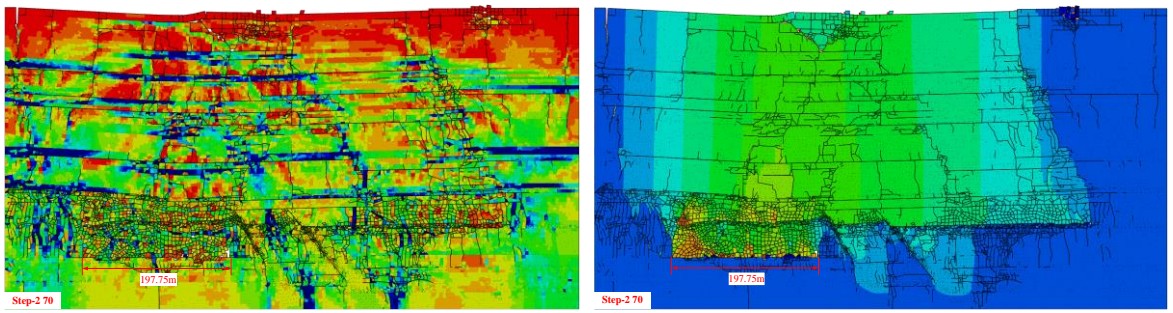

(**d**) Mining advance 197.75 m.

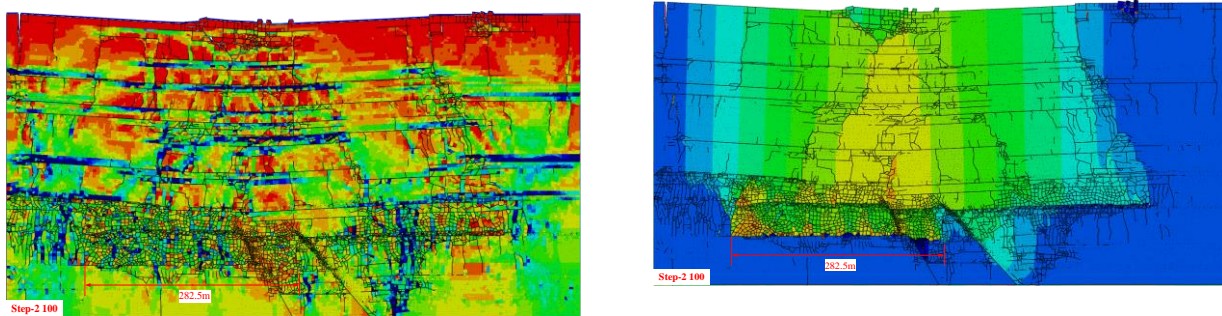

(**e**) Mining advance 282.5 m.

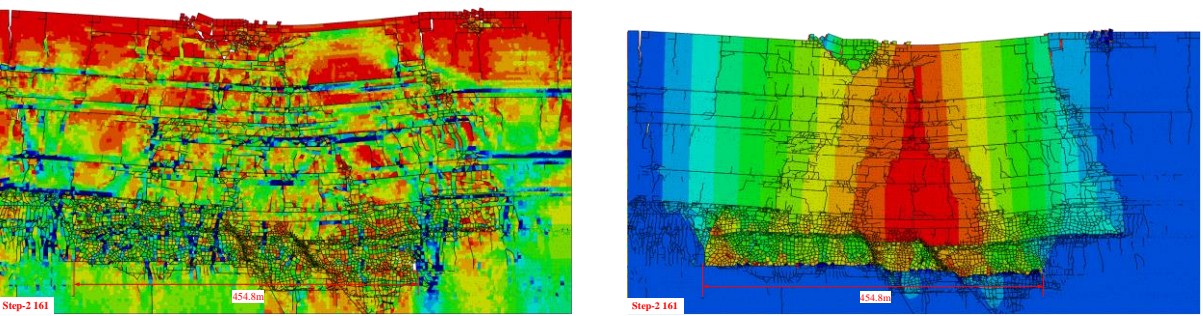

(**f**) Mining advance 454.8 m.

**Figure 9.** *Cont.*

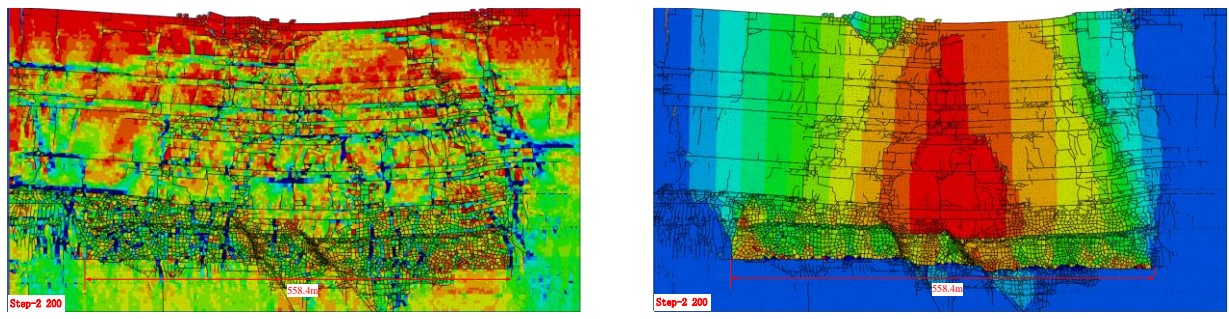

(**g**) Mining advance 558.4 m.

**Figure 9.** Overburden rock fracture distribution, Mises stress, and vertical displacement field.

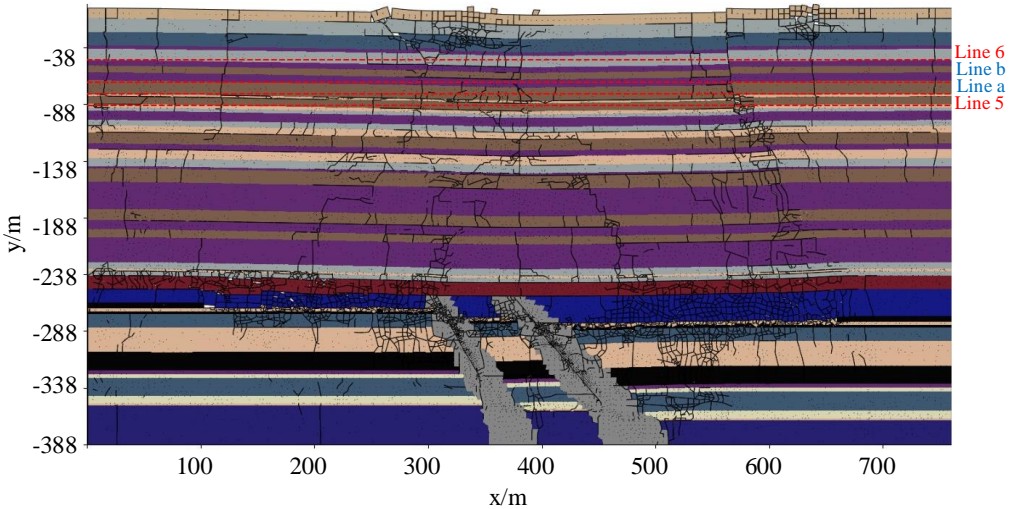

(**a**) The location of the encrypted survey line.

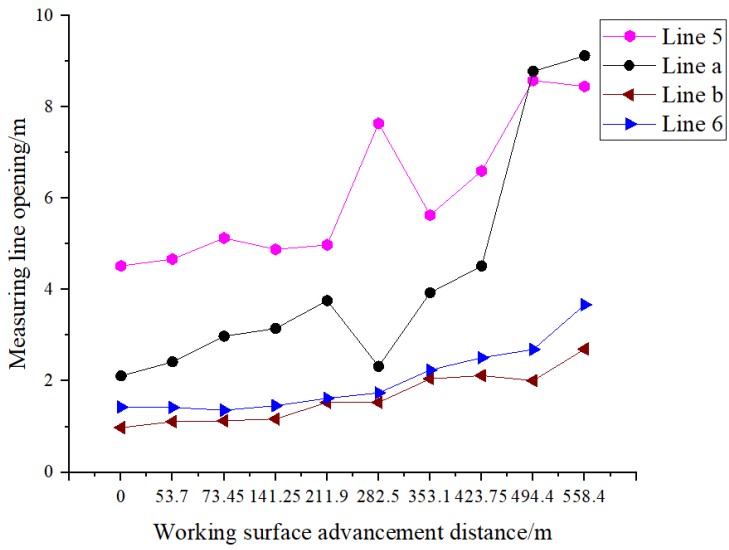

(**b**) Curve chart of crack width at survey lines.

**Figure 10.** Layout of measuring lines and crack opening width mining advance.

In addition to the fracture aperture mentioned above, fracture density is also a key factor affecting the water conductivity of rock strata. The crack density of the 2 # roof determines the amount of water inflow. The fracture density in the advancing direction of

the working face is statistically analyzed, and the fracture density characteristics during the working face approaching the fault–crossing the fault–far away from the fault are obtained (Figure 11).

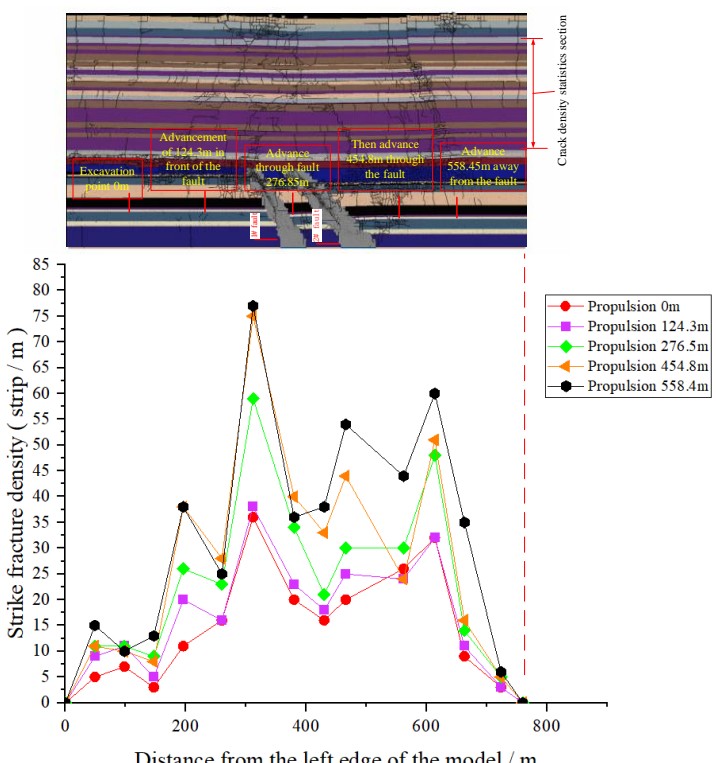

**Figure 11.** Strike fracture density characteristic curves at different working face advancing distances.

It can be seen from Figure 11 that the roof crack density shows a four-stage evolution trend with the advancing distance of the working face: In the first stage, the advancing distance of the working face increases from 0 m to 124.3 m (that is, the working face has not yet crossed faults), the fissures formed after the mining of the 2 # coal seam are compacted due to the disturbance of 5 # coal seam mining, and some new fissures are produced at the same time. Under the influence of the two, the growth is not obvious. In the second and third stages, the advancing distance of the working face is from 124.3 m to 276.85 m. At this time, the working face passes through faults for the first time and the second time, respectively, and it can be observed that the fracture density increases obviously at this time. When the working face pushed through the fault at 30 m (Figure 9d), due to the poor mechanical properties of the fault, and under the influence of the superposition effect of mining stress of 2 # coal and 5 # coal, the stress concentration appeared on the hanging wall of the fault and was larger than the conventional position. After passing through the fault zone, the fault zone is broken and the stress is released. The mudstone and siltstone of the 5 # coal roof are broken along the fault. The cracks of the fault itself and a large number of newly generated cracks extend upward and propagate to the upper roof, and the density of the strike cracks increases obviously. When the mining advance is 276.85 m (Figure 9e), the hanging wall and footwall of the fault have stress concentration and are larger than the conventional position. When the working face is 454.8 m (Figure 9f). The mudstone and siltstone of the roof have been broken, the fault zone is broken, the stress is released, the cracks of the fault itself and a large number of newly generated cracks extend upward and propagate to the upper roof, and the density of the strike cracks increases obviously. In the fourth stage, the advancing distance of the working face is from 276.85 m to 558.4 m. The overlying rock in the goaf is gradually caving and compacting, and some

new fissures are produced at the same time. The overall growth trend slows down and the local density increases.

It can be seen from Figures 8 and 9 that the overburden fracture zone near the fault zone showed different characteristics from other areas. In the process of continuous advancement of the working face, due to the concentration of stress in the geological structure, new rock fractures formed. These cracks and fissures formed a fracture zone, which made the fracture zone near the fault denser (Figures 10 and 11). The overburden fracture zone near the fault zone was generally more continuous, because the existence of the fault zone provided continuous stress and promoted the formation and growth of the fracture zone (Figures 9 and 12).

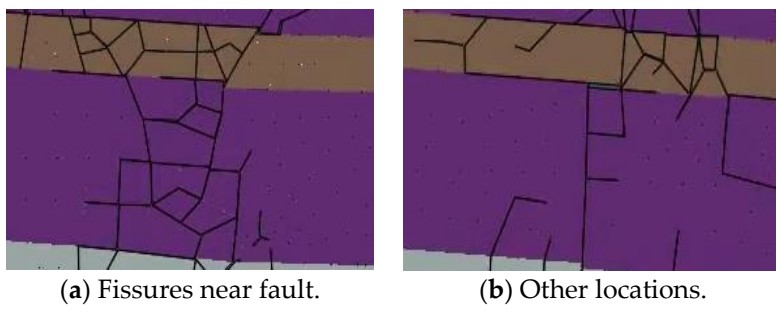

(**a**) Fissures near fault.          (**b**) Other locations.

**Figure 12.** Comparison between the overlying rock fracture zone near the fault zone and other locations of fractures.

In summary, in the process of coal seam mining, sandy mudstone, due to the characteristics of the overlying rock layer, is more likely to be affected by mining-induced stress, resulting in the rock body fracturing across the fault, leading to the further growth of the original fissures and the generation of new fissures, as well as possible roof water-related accidents. The existence of fault zones leads to local rupture, cracking, and fissure formation in the roof rock body. Therefore, attention must be paid to the location of faults in the mining process. For instance, appropriate support and reinforcement measures for the fault surface need to be taken to reduce the impact of stress concentration and ensure the stability of the working face and the roof plate. Finally, from the analysis of our case study, it can be concluded that the height of water-conducting fissures was between 216 m and 226 m.

*5.4. In Situ Experiment*

In order to verify the results of the numerical simulation of water-conducting cracks in overlying rock, in situ experiments were carried out to analyze the loss of borehole flushing fluid. The experimental parameters were set as follows: the elevation of the hole opening is 430 m, elevation of coal roof 5# is 1085 m, positioning of the hole from the cut hole is 180 m, and depth of the hole is 300 m. The borehole was injected with water in certain sections after mining, and the water loss was observed.

From the post-mining leakage test involving the hole, it was found that the first instance where there was no water return in the hole was when the hole depth was 140 m, and the maximum leakage amount was 0.0493 L/s. Then, there was no water return again at 141.09 m. The leakage per unit of time at this location was 0.0565 L/s, and the maximum leakage per unit of time per unit of feed was 0.1357 L/s. Below the hole depth of 141.09 m, the borehole did not return water, indicating the propagation of cracks in the hole wall. Both the loss of drilling fluid and the loss of unit footage per unit time increased significantly, unequivocally indicating that the mining of coal seam 5# compromised the integrity of the original rock formations in this section, resulting in the formation of layer gaps and fissures. The experimental results prove that the height of the water-conducting fracture zone after the mining of Yushupo coal seam 5# is 205 m. The validity of the above

numerical simulation results is thus further verified. The specific changes in leakage are shown in Figure 13.

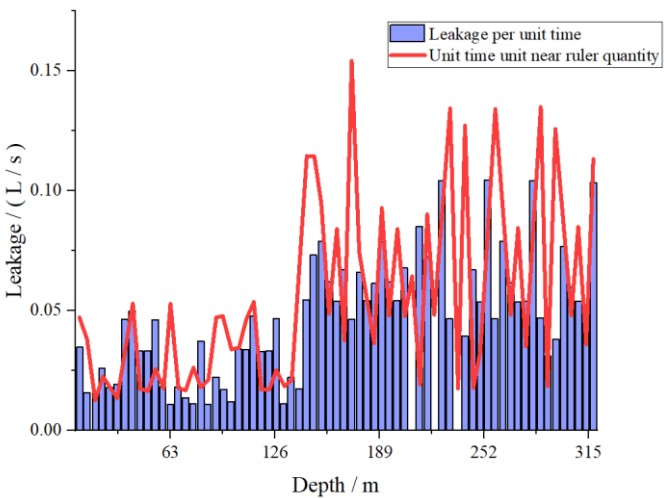

**Figure 13.** Leakage evolution of flushing fluid in borehole after coal mining.

## 6. Discussion

In this study, the constitutive relation of the continuous deformation–discrete motion of rock was established. On this basis, the FDEM finite–discrete element program was compiled to realize the process of transforming the continuum into random discrete blocks, and the VUSDFLD algorithm subroutine was compiled to realize the numerical simulation of the water-conducting fracture evolution of overburden rock during fault crossing of extra-thick coal seams. Based on the above procedures and the results of in situ experiments based on the flushing fluid consumption method, the propagation of water-conducting fissures in the overlying rock layer of extra-thick coal seams across faults was studied. The above results were initially applied in the Ningwu mining area. We found that due to the low fracture toughness of sandstone, this formation is more likely to fracture under the stress induced by the continuous mining of multiple extra-thick coal seams. The height of the original overlying rock fissures finally stabilized at 216~226 m (Figure 10). Due to the greater load on the overlying rock layer in the fault zone, the cracks in the overlying rock layer at the working face were significantly larger than conventional ones (Figures 11 and 12a,b). Compared with previous studies [16,20–26], this research finding is of great significance. This study was conducted in the context of the mining of extra-thick coal seams passing through a double fault; given that high stress concentration in extra-thick coal seams, stress in multiple coal seams, and stress caused by double faults intensify the water inrush risk, our research results are even more significant. We validated the accuracy of the numerical results in terms of modeling the characteristics of the mechanical response of rock strata to the transformation from quasi-continuous media into discrete bodies due to continuous coal mining, simulating the continuous mining of multiple coal seams, modeling the discrete–continuous transformation of rock media, and modeling the continuous generation and release of mining stress in this process. In addition to the above, we verified the original experimental results by using the flushing fluid consumption method. To the authors' knowledge, this is the first time that the finite element–discrete element method (FDEM) algorithm and the continuous mining subroutine (VUSDFLD) are combined to study the propagation law of water-conducting fractures under the engineering geological conditions of multiple extra-thick coal seams passing through double faults.

So far, most studies on water-conducting fractures in overlying rock mass have been based on the continuous and discrete medium hypotheses [12,23,25,29]. Employing the finite difference method (FDM), finite element method (FEM), and discrete element method (DEM) produces acceptable simulation results in roofs with obvious plastic characteristics;

joints that are developed, interconnected, or completely broken; and cracks that are interconnected. However, using the FEM and FDM only allows one to obtain the deformation and plastic zone of the overlying rock layer, not roof fracture and collapse numerical results. When applying the DEM, it is usually assumed that all joints run through the entire numerical model in a "block distribution", often without considering the intact rock mass. Fractures presenting plastic and non-penetrating characteristics can easily lead to large deviations between numerical results and actual measurement results. When a working face at multiple extremely thick seams passes through double faults, the coal seam roof, especially in the fault zone, undergoes a significant transformation from continuous media into randomly shaped discrete bodies.

More importantly, during this process, quasi-brittle rock exhibits tension–shear mixed ductile fractures, and natural and newly emerged structural planes exhibit separation, compression, and shear friction coupling responses (Figure 14). The above continuous-to-discrete conversion process is difficult to model by only using the FEM, FDM, or DEM [30]. Although some scholars have proposed research based on the FDEM framework [40,41], there are few studies on the laws of water-conducting crack propagation in overlying rock strata. Related studies also present a common limitation, that is, to only analyze simple geological structures. At the same time, in the above studies, coal seams are mostly regarded as discrete-block groups; after reaching a certain time threshold, discrete-block groups are instantly deleted for the purpose of working face advancement. Coal seam mining is a continuous process of generating and releasing mining stress, and stress is the root cause of coal and rock damage.

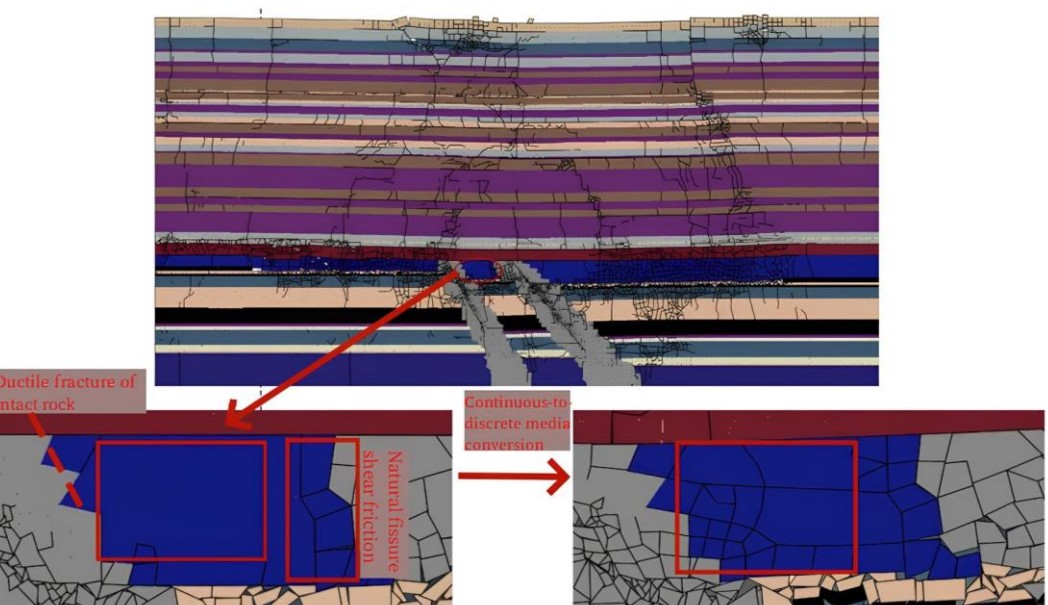

**Figure 14.** Rock mass continuous–discrete conversion.

In this study, we established a numerical model for the ultra-thick coal seams in Yushupo, which cross a double fault, based on a numerical model of water-conducting fissure propagation in the overlying rock layer and the continuous deformation–discrete movement constitutive relation of the rock mass. As shown in Figure 5, we numerically modeled the process of the transformation of the double fault and its overlying rock layer from (quasi-)continuous media into discrete bodies and the formation of water-conducting cracks in the overlying rock layer due to stress from the continuous mining of multiple extra-thick coal seams; further, we inferred the propagation law of water-conducting cracks in overlying rock mass. These models, constitutive relations, and subroutines provide a new basis for numerical simulation in coal seam mining and research on the propagation

laws of water-conducting fissures in overlying rock mass under conditions of multiple extremely thick coal seams crossing double faults.

To reduce the development of mining cracks, we can have the following ideas: Mining coal seams uses selective mining methods [42]. That is, the rock is left in the ground after coal mining, which alters the pressure distribution in the formation and provides support to slow down the movement of the formation, thus reducing the formation and expansion of cracks. Coal waste rock filling also increases the overall stability of the formation, reduces surface settlement and collapse caused by mining, and reduces the development of hydraulic fracturing. Leaving the rock in the ground can plug the cracks, change the path of water flow, prevent water from accumulating in the mining area, and prevent disasters such as surface water infiltration. At the same time, we can further study the impact of backfill materials and methods on crack development, etc., to enhance the fill-ability of rock.

However, it is necessary to acknowledge the limitations of this work. Simulation has high requirements on parameters, and the accuracy of parameters will affect the results of numerical simulation. We must pay attention to the fact that in actual engineering applications, the mechanical parameters of the simulation should be obtained as accurately as possible. In addition, the statistics of the crack opening in the simulation are actually roughly counted based on each crack, and errors are inevitable. Designing a crack-opening statistical algorithm can effectively solve this problem. These are all worth considering for further research and optimization.

## 7. Conclusions

(1) With the established continuous rock deformation–discrete motion constitutive relation and the corresponding FDEM, the VUSDFLD multi-coal seam continuous mining algorithm subroutine was compiled to achieve the continuous mining of multiple coal seams and accurately reproduce the formation process of water-conducting fracture zones under the conditions of extra-thick coal seams crossing faults.

(2) Under simulation conditions, the height of the water-conducting cracks in the overlying rock layer caused by the continuous mining of extra-thick multiple coal seams was in the range of 216~226 m, and the fracture–mining ratio was 14.1, presenting a deviation from the in situ experimental results of borehole flushing fluid loss of 5%~10%.

(3) Compared with non-structural locations, overlying rock strata in fault zones are subject to greater loads under the stress induced by extra-thick coal seam mining, resulting in a significant increase in the density and width of overlying rock cracks in the upper roof, exacerbating the risk of roof water inrush.

**Author Contributions:** Software, L.M.; Validation, L.M.; Formal analysis, H.L.; Investigation, H.L.; Resources, H.L.; Data curation, L.M.; Writing—original draft, L.M.; Writing—review & editing, H.L.; Visualization, L.M.; Supervision, L.M.; Funding acquisition, H.L. All authors have read and agreed to the published version of the manuscript.

**Funding:** This work was supported by the National Natural Science Foundation of China (No. 42102310), and Fundamental Research Program of Shanxi Province (No. 20210302123106).

**Institutional Review Board Statement:** Not applicable.

**Informed Consent Statement:** Not applicable.

**Data Availability Statement:** Data are contained within the article.

**Acknowledgments:** The authors would like to thank anonymous referees for their careful reading of this article and valuable suggestions.

**Conflicts of Interest:** The authors declare no conflict of interest.

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
