# Peer review of "Finite–Discrete Element Method Simulation Study on Development of Water-Conducting Fractures in Fault-Bearing Roof under Repeated Mining of Extra-Thick Coal Seams"

_sustainability, doi:10.3390/su16125177_

Round 1
Reviewer 1 Report
Comments and Suggestions for Authors
This article takes the development height of water-conducting fissures in the mining overburden under the condition of extra-thick multi-coal seams and fault group. The mechanical constitutive relationship of deformation-fracture-discrete block interaction of the mining overburden was established. The corresponding finite element-discrete element (FDEM) numerical program was compiled, and the K_VUSDFLDBYSTEM multi-coal seam continuous mining algorithm subroutine was compiled. The author has conducted a lot of research work, which can provide certain theoretical support for water-containing coal mining and the controlling of water inrush from roof under extremely thick coal seams. From the overall content, there are good innovative ideas, but there are still the following problems:
1. In the second part, the author spends a lot of space introducing FDEM coupling theory. Although Figure 3 shows the numerical process of FDEM, since the understanding of subsequent numerical simulation results is very important, a more detailed basic principle of the numerical process needs to be provided. It is recommended to add this content to Section 4.1.
2. The author’s wording on “extremely thick and multiple coal seams” in the title “FDEM Simulation Study on the Evolution of Water-conducting Fissures in Overlying Rocks Overlying Fault Groups in Working Faces with Extra Thick and Multiple Seams” lacks scientificity and universal applicability.
3. There is too much description of the secondary development of subroutines in the article, and it is recommended to refine it.
4. In reviewing this manuscript, it became necessary to improve the details of the figures. For example, in Figure 4, etc., text outside the wireframe limits legibility.
5. In conclusion (2), the statement of the conclusion lacks rigor. By simulating the formation process of water-conducting cracks in the overlying rock during the mining of extra-thick and multiple coal seams, the author obtained the height of the water-conducting cracks in the overlying rock caused by repeated mining disturbances of extra-thick and multiple coal seams. The height is 216m~226m, which is consistent with the in-situ experimental results of the loss of borehole flushing fluid. Deviation of 10%. The height of water-conducting cracks in the overlying rock obtained through simulation is a numerical range, while the deviation value is a specific numerical value. Please explain the reasons for this expression.
Minor editing of English language required.
Reviewer 2 Report
Comments and Suggestions for Authors
The paper explores the formation of water-conducting fractures in the roof rock of coal mines, especially when mining extra-thick coal seams with faults. Using a combination of finite and discrete element methods (FDEM) in numerical simulations, the study aims to predict and understand how these fractures develop and propagate. The research is focused on a specific coal mine, Yushupo Coal Mine, and evaluates how repeated mining activities impact the roof stability and fracture formation. Key findings include the identification of stress concentrations around faults, leading to more pronounced fracturing, which can help in managing roof water inrush disasters in similar mining conditions.
The manuscript has practical applications and offers significant theoretical insights for future research. It can be accepted for publication once the following corrections are addressed.
Comment 1.
Use separate words (keywords) not a combination of several words in the abstract.
Comment 2.
The Introduction section presents a thorough literature review, primarily emphasizing research conducted in China. However, it falls short in analyzing studies from other countries. This aspect requires substantial enhancement to offer a more balanced perspective.
Comment 3.
Motivation of the paper is not clear.
Comment 4.
It would be helpful if the authors provided some context to highlight the significance of conducting this study.
Comment 5.
Please consider the suggested research in your paper when enhancing the literature review (enhanced review of non-China authors only are more than welcome). I believe they are worth considering in your paper.
Vu, T.T (2022). Solutions to prevent face spall and roof falling in fully mechanized longwall at underground mines, Vietnam. Mining of Mineral Deposits, 16(1), 127-134. Doi:10.33271/mining16.01.127
Rak, Z., Stasica, J., Burtan, Z., & Chlebowski, D. (2020). Technical aspects of mining rate improvement in steeply inclined coal seams: a case study. Resources, 9(12), 138. https://doi.org/10.3390/resources9120138
Comment 6.
What are the limitations of your research?
Comment 7.
Lines 95-96. “The VUSDFLD algorithm subroutine was compiled to realize multi-coal seam continuous mining” Why VUSDFLD algorithm?
The aim and the tasks must be highlighted at the end of the Introduction section.
Comment 8.
What is the novel in the section 2? Is it so important to provide it in the paper? The basic knowledge is important but it take a large part of the paper.
Comment 9.
Mechanical parameters should have full explanation in the first row of the table 1.
Comment 10.
There are several times Mpa instead of MPa in the table 1.
Comment 11. Line 214. σn= 3,8MPa. Use dot instead of comma. Check and correct across the text.
Comment 12. Subsection 5.1 should be titled as “Study area”
Comment 13. Figure 2d. should be MPa instead of Mpa
Comment 13.
Line 505. Could you provide more details on how the combination of the FDEM algorithm and the continuous mining subroutine (VUSDFLD) specifically enhances the study of water-conducting fracture propagation in comparison to previous methods?
Comment 14. What about the influence of leaving rock underground after coal mining? Can such actions impact the development of water-conducting fractures in fault-bearing roofs? Please consider mentioned below research in your study as it can affect further research direction.
Malashkevych, D.; Petlovanyi, M.; Sai, K.; Zubko, S. Research into the coal quality with a new selective mining technology of the waste rock accumulation in the mined-out area. Min. Miner. Depos. 2022, 16, 103-114. https://doi.org/10.33271/mining16.04.103
Comment 15. Please provide a brief description of potential directions for further research.
Comment 16. References. There are no citing papers published in 2024 and 2023, and only two papers published in 2022 and three published in 2021 are analyzed. The relevancy of the research is under question.
Comment 17. Overall, I must acknowledge that a very good study was conducted, and I will recommend your paper for publication after careful revision.
Reviewer 3 Report
Comments and Suggestions for Authors
The article applies a new method of FDEM secondary development to the simulation of the evolution laws of multi-coal seam fissures. From the overall content, it has certain innovative ideas, but there are still some problems in the article:
1. How are the overlying rock structural planes expressed, and how their spacing is determined?
2. The title of the article is too long and needs to be shortened.
3. In terms of the research content in the article, the author only studies the development height of water-conducting fissures in the mining overlying rock under the condition of the 5# extra-thick coal seam crossing the fault group, while the title states it as "thick coal seam working face".
4. In the manuscript legend, there is a problem that the text exceeds the line frame and is unclear, as shown in Figure 2, etc. The author should check it carefully.
5. In the simulation of ultra-thick coal seams crossing fault groups, what method is used for the working face to cross faults? Specific engineering methods should be presented.
6. It is more reasonable to use "water-conducting fissures" in the article as "coal mining coal seam overlying rock fissures". Is this a confusion of concepts?
Round 2
Reviewer 2 Report
Comments and Suggestions for Authors
While I appreciate that revisions have been made, it is challenging to identify where my recommendations have been addressed in the revised manuscript. Furthermore, it appears that several suggestions have not been incorporated, as reflected in the manuscript and your responses. My suggestions were not only intended to inform but also to guide improvements in the paper.
Please provide a detailed response to each of my comments, specifying the exact lines in the manuscript where changes were made based on recommendations.
A literature review is a critical issue. You have added only 3 recently published papers. It is not a serious approach. My recommendation to enhance this part was ignored. Target suggestions were also not included.
Round 3
Reviewer 2 Report
Comments and Suggestions for Authors
Dear authors, I wish to express my utmost satisfaction with the revisions you have implemented. I offer my heartfelt congratulations to you for your diligent work. Perfect!